# Relationship between Resting and Recovery Heart Rate in Horses

**DOI:** 10.3390/ani10010120

**Published:** 2020-01-11

**Authors:** Arno Lindner, Martina Esser, Ramón López, Federico Boffi

**Affiliations:** 1Arbeitsgruppe Pferd, 52428 Juelich, Germany; 2Centro de Fisiología y Fisiopatología del Equino Deportivo, Facultad de Ciencias Veterinarias, Universidad Nacional de La Plata, Buenos Aires 1900, Argentine

**Keywords:** endurance, exercise, health, lactate, welfare

## Abstract

**Simple Summary:**

Veterinary check points are established at certain distances within an endurance race to ensure the health of the competing horses. In the vet checks horses are stopped and examined. Heart rate (HR) is one of the parameters controlled. The horses can continue racing after reaching a pre-defined and arbitrary HR only, providing all other parameters are in good order too. The results of this study show that the time until the pre-defined HR is reached is shorter when the resting HR of a horse is lower. The study shows that the lower resting HR is not related to a higher endurance. Therefore, pre-determined arbitrary HR recovery values may not allow for fair competition during endurance racing. Instead, HR recovery tests based on individual HR should be introduced.

**Abstract:**

In endurance racing the heart rate (HR) of horses in the veterinary gates has to reach a maximum set to continue racing. There is no literature on the relationship between resting HR (HRresting) and HR after exercise (HRrecovery). This relationship was examined in seven horses and the results were related to their v_4_ (speed at which the blood lactate concentration is 4 mmol/L). Horses were submitted to an exercise test to determine v_4_. Thereafter, horses were exercised on a treadmill in randomized order for 10 and 60 min at different speeds. HR was measured before exercise and several times until 30 min of recovery. The relationship between HRresting and HRrecovery was significant in 16 out of 35 comparisons. There were no significant relationships between the v_4_ of the horses and their HRresting and between v_4_ and HRrecovery after 10 min of exercise, regardless of the speed of exercise, with one exception. The relationship between the v_4_ of the horses and their HRrecovery after 60 min of exercise was significant in the fifth minute after exercise at 3.5 m/s only. Conclusion: Because HRresting and HRrecovery are often related, pre-determined arbitrary HRrecovery values may not allow for fair competition during endurance racing.

## 1. Introduction

There are many exogenous and endogenous factors affecting the resting heart rate (HRresting) in horses [1,2]. According to several studies resting HR (HRresting) can be lowered through training [3,4,5,6,7] although there are many other studies that did not report such an effect [8,9,10,11,12,13,14,15]. The type of training program applied is likely to be an explanation for the differing results. Genetics could also play a role [16,17]. A study in humans suggests another dimension requiring investigation in horses [18]. They found that multiple genes contribute to genetic variation in the adaptation of submaximal HR after exercise (HRexercise) to regular physical activity. Thus, chances are that a change in HR will not occur or occur to a lesser extent in an individual with no or fewer certain alleles, and therefore the response of HR parameters to training might be quite individual.

In endurance racing horses are stopped several times during a race for a veterinary inspection at so-called vet gates. For a horse to continue competing, its HR must be at or below a fixed value within 20 min of arrival. In general, the upper limit for an 80-km race or longer is a HR of 64 beats/min [19]. The Veterinary Commission may change the fixed HR or the recovery time before or during the competition to adjust to particular conditions. The time between arrival at each vet gate and the start of the veterinary inspection is counted as part of the overall riding time. Thus, a fast recovery HR is very important for success in endurance events. Any horse deemed unfit to continue (due to lameness or HR not recovering as expected, for example) is withdrawn from the event. The fixed HR rule has developed a life of its own because HR is only one criterion of the health status of a horse during an endurance race [20]. Setting HR limits has led to the selection of horses with a low HR at rest and to trying to reduce it with drugs, manual manipulation, and other means [20,21]. However, until now there has been no evidence that a lower HR at rest signifies better endurance or overall fitness, while some authors have concluded that there is no such relationship [8,9,10,11]. No literature on the relationship between resting HR (HRresting) and HR recovery (HRrecovery) was found. Thus, the objective of this study was to confirm or refute the hypothesis that there is a positive relationship between HRresting and HRrecovery in horses.

## 2. Materials and Methods

### 2.1. Horses

Seven horses (five purebred Arabian and two Arabian-cross; two aged 2 years, two aged 3 years, one aged 4 years, one aged 5 years, and one aged 8 years; four mares, one stallion, and two geldings) adapted to exercising on a treadmill and being prepared for endurance racing or already competing were used. All procedures were approved by the Bioethical Committee of the University of La Plata, Argentina.

### 2.2. Experimental Design

In the morning, 2 h after feeding, horses run every second day on a treadmill at one of the speeds selected for 10 or 60 min (Mustang 2200, Kagra AG, Fahrwangen, Switzerland). First the horses were warmed-up always for 1 min at 1.5 m/s and for 4 min at 3.5 m/s on a 0% slope. Thereafter, the horses were exercised in a randomized order for 10 min at one of the following speeds on a 3% slope: 1.5, 3.5, 5, 6, and 7 m/s (Table 1). Two runs of 60 min each at 3.5 m/s and 5 m/s followed the series of 10-min runs. The order of these runs on different days was also randomized (Table 1). After all exercises the horses continued walking on the treadmill at 1.5 m/s and 0% slope for 10 min and were then walked to their paddocks where they remained until a total recovery time of 30 min was reached.

### 2.3. Heart Rate Measurement

Commercially available HR meters were used for the HR measurements (Polar S610, Polar Electro, Kempele, Finland). The HR meters were attached to the thorax of the horses according to the manufacturer’s instructions. The HR was recorded at 5-s intervals. Gel was applied to improve the conductivity of the HR signal between the electrodes and the skin of the horses (F7 Gel, Gel Conductor Classic, Laboratorios FABOP, Buenos Aires, Argentina). The data recorded by the HR meter were transferred to a computer through an interface (Polar USB IR Interface, Polar Electro, Finland) for analysis on a computer with special software (Polar Equine SW, Version 4.02.036 H, Polar Electro, Finland).

The resting HR (HRresting) was determined on three consecutive days in the morning 30 min before feeding. After placing the HR meters on the thorax of the horses they were left alone in their paddocks and all disturbances avoided. The HR was recorded for 15 min and the section of the recording with steady signals was used to calculate the mean HR, representing HRresting.

The HR after exercise (HRrecovery) was recorded for 30 min after ending the 10- and 60-min exercises (HRrecovery10 and HRrecovery60, respectively). The means of the values recorded during the minute before the 1, 5, 15, and 30 min of recovery were used to determine HRrecovery10 and HRrecovery60. In addition, the time between the end of exercise and when the HR of 64 beats/min was reached (HR64), was determined.

### 2.4. v_4_ (Speed at Which the Blood Lactate Concentration is 4 mmol/L)

A standardized submaximal exercise test (SET) was performed by each horse on the treadmill at the beginning of the study to determine the blood lactate running speed relationship (BLRS). Blood samples were drawn from the jugular vein prior to SET, but after warm-up. Warm-up consisted of 5 min at 1.7 m/s and 5 min at 4.0 m/s on a 0% slope.

The SET on the treadmill inclined at 6% consisted of 5 min at each speed, starting at 4.0 m/s with subsequent steps increasing incrementally by 0.5 m/s. Between each step, the treadmill was stopped for 60 s, blood samples were drawn (within 15 s) by puncture of the jugular vein into Li-heparinized evacuated tubes (Becton Dickinson, Heidelberg, Germany) and blood Lactate (LA) was determined immediately (Accusport™; Roche Diagnostics, Mannheim, Germany). When the horse reached a speed at which the blood LA concentration was at or above 4 mmol/L, the SET ended. A BLRS curve was generated using exponential regression analysis and used to calculate the horse’s speed at which the LA concentration reached 4 mmol/L (v_4_) [22].

### 2.5. Statistical Analysis

All analyses were run on Statview 5.0 (SAS, Cary, NC, USA). All data are expressed as mean ± standard deviation (SD). Analysis of variance for repeated measurements was applied to determine the effect of the speed of exercise on the HRrecovery. When a significant F ratio was achieved with the level of significance fixed at *p* < 0.05, post-hoc comparisons were carried out via Fisher’s least significant test to locate specific significant differences among exercises. The Pearson coefficient of correlation was calculated to examine the relationship between HRresting and the age of the horses, HRresting and HRrecovery, HRresting and Time64 (the time when a HR of 64 beats/min), as well as HRresting and v_4_. *p* < 0.05 was set as the limit to denote significance.

## 3. Results

Mean HRresting was 31.7 ± 5.3 beats/min. Mean HRresting did not depend on age (*p* > 0.05). The mean HRresting and HRrecovery10 values of the horses are shown in Table 2, and those of HRrecovery60 in Table 3.

### 3.1. Exercise for 10 min

HRrecovery10 was higher in the first and fifth minute after exercise at speeds at 5 m/s and higher than after lower speed exercises (*p* < 0.01) but not in the 15 and 30 min after exercise (*p* > 0.05; Table 2).

The time when HR64 was reached after 10 min of exercise depended on exercise speed (*p* < 0.01; Table 4).

### 3.2. Exercise for 60 min

The mean HRrecovery60 differed between speeds during the first minute only (*p* < 0.05; Table 3). The mean HRrecovery60 at 5, 15 and 30 min after exercise did not differ between the two exercise speeds (*p* > 0.05).

The mean times to reach HR64 after 60 min of exercise were similar for both speeds of exercise (*p* > 0.05; Table 4).

### 3.3. Relationships between HRresting and HRrecovery

HRresting and HRrecovery10 were more often significantly correlated when exercise speed was 1.5 m/s, 3.5 m/s and 5 m/s and less often when the speed was 6 m/s and 7 m/s (Table 5).

The time taken to reach 64 or less beats/min was correlated with HRresting after 10 min exercise at 3.5 m/s and 5 m/s only (Table 5).

Horses with a higher HRresting showed higher HRrecovery60 values 15 and 30 min after exercise (*p* < 0.05 to < 0.001; Table 6). HRresting was not correlated with HR64 after running for 60 min at 3.5 or 5 m/s (*p* > 0.05; Table 6).

### 3.4. Relationships between v_4_ and HRresting, v_4_ and HRrecovery

The mean v_4_ of the horses was 5.58 ± 0.32 m/s (range 5.1–6.2 m/s). There was no significant relationship between the v_4_ of the horses and their HRresting (*p* > 0.05).

There were no relationships between v_4_ and the different values of HRrecovery10, regardless of the speed of exercise (*p* > 0.05). The exception was a positive relationship between v_4_ and the HRrecovery10 in the first minute after exercise at 1.5 m/s (*p* < 0.05; r^2^ = 0.56).

The relationship between the v_4_ of the horses and their HRrecovery60 was significant and positive in the fifth minute after exercise at 3.5 m/s only (*p* < 0.01; r^2^ = 0.80).

The relationship between the v_4_ of the horses and their HR64 was positive after horses had run at 3.5 m/s for 10 min and 60 min (both *p* < 0.05; r^2^ = 0.60 and r^2^ = 0.68, respectively), but not when horses had run for the same times at other speeds (*p* > 0.05).

## 4. Discussion

The results of this study indicate that it is reasonable to assume that there is a positive relationship between HRresting and HRecovery. This would mean that horses with a lower HRresting could leave the vet gates during an endurance competition earlier than horses with a higher HRresting because they would reach the HRrecovery value fixed beforehand by the racing authorities within a shorter time, giving them a competitive advantage. Obviously the relatively small number of horses participating in the study and the short duration of exercise in comparison to the duration of endurance races of 80 and more km precludes generalization of this assumption. Longer races will produce much larger thermoregulatory increases and fluid and metabolic shifts in the horses than in this study [23,24] and these effects may affect the relationship. These effects might also be the cause of the fewer significant relationships observed between HRresting and HRrecovery when the horses had run at 6 m/s and 7 m/s compared to the lower speeds of exercise.

On the other hand, the speeds of 3.5 m/s to 7 m/s, used to exercise the horses in this study, correspond with the range of winning speeds in endurance races worldwide [25] and the standardization of all procedures allow confirmation that there is a positive relationship between HRresting and HRrecovery in horses. This relationship has long been taken for granted in practice and is the reason for the worldwide search for horses with low HRresting to compete in endurance races.

There was no relationship found between v_4_ of horses and their HRresting and very few ones between v_4_ and HRrecovery in this study. v_4_ is the parameter that has most often been shown to be associated with the competitive performance of horses [26], including in endurance racing [27,28,29,30]. Thus, there appears to be no scientific evidence for the fixed HRrecovery system in endurance racing. This system may not only be preventing fair competition conditions for all horses, but may be compromising their health and welfare because the HR of horses with lower HRresting may be able to recover sufficiently to continue competing despite their health being compromised already.

Based on the results of this study it would be much better to define the individual HRrecovery for each horse before competition and apply this value in the vet gates during competition. The individual HRrecovery could be determined during the veterinary examination of a horse to determine if it is fit to compete. Individual HRrecovery indices have been proposed by others already [31,32].

The speed of exercise had an effect on HRrecovery rate in the first 5 min after exercise only. Thereafter the decline in the rate of HRrecovery was the same. This result validates the racing strategy of reducing the running speed before entering a vet gate to increase the likelihood of reaching the fixed HR value earlier. The time taken to reach HR64 also depended on the duration of exercise, taking longer when the horses exercised for 60 min than for 10 min. These effects of exercise speed and duration on HRrecovery have been described previously [9,10,33,34,35,36].

The management of the horses following exercise could have affected their HRrecovery rates. During the first 10 min of the recovery phase the horses were walked on the treadmill. Thereafter, they were walked to their paddocks within 5 min and left loose in them for the remaining time until completing the total of 30 min set for recovery observation. It is known from other studies that HR recovers more rapidly when horses are walked, and even more rapidly when trotted instead of kept standing [37,38]. Thus, it is likely that the HRrecovery rates of the horses in this study would have been somewhat higher after the 15th minute if they had been walked until the end of the recovery period of 30 min.

The HRresting of the horses in this study did not depend on their age. Such a relationship has been reported previously [39,40,41], with the exception of Younes et al. [42]. These discrepancies might be due to breed and age differences of the horses used in the studies.

## 5. Conclusions

In this investigation positive relationships were found between HRresting and HRrecovery at many different times after exercise. There are some caveats for interpreting these results because the effects of longer duration exercise on this relationship, as in endurance racing, cannot be foreseen. However, fixed HRrecovery values may not allow fair competition because it is likely that there is no relationship between HRresting and endurance and there may be none between HRrecovery and endurance either. Fixed HRrecovery values could even impair the health of horses during endurance racing. Instead, HRrecovery tests based on individual HR before and after a defined exercise should be introduced.

## Figures and Tables

**Table 1 animals-10-00120-t001:** Order of exercises for each horse at the different speeds (m/s) for 10 and 60 min duration.

Horse	Order of Exercises						
	First	Second	Third	Fourth	Fifth	Sixth	Seventh
	10 Min Duration					60 Min Duration	
1	1.5	3.5	6	7	5	3.5	5
2	5	7	3.5	1.5	6	3.5	5
3	7	1.5	5	6	3.5	5	3.5
4	3.5	6	1.5	5	7	3.5	5
5	3.5	6	1.5	5	7	3.5	5
6	1.5	3.5	6	7	5	5	3.5
7	6	5	7	3.5	1.5	5	3.5

**Table 2 animals-10-00120-t002:** Recovery heart rate at different times after 10 min of exercise at different speeds (mean ± SD; 7 horses).

Time (min)	Speed (m/s)				
	1.5	3.5	5	6	7
1	57.6 ± 15.1 ^a^	74.9 ± 12.1 ^a^	93.7 ± 18.6 ^b^	122 ± 16.9 ^b^	126 ± 14.5 ^b^
5	38.6 ± 9.9 ^a^	44.3 ± 10.7 ^a^	56.1 ± 13.7 ^b^	73.3 ± 10.5 ^b^	75.9 ± 12.9 ^b^
15	39.1 ± 10.9 ^a^	41.7 ± 8.0 ^a^	44.6 ± 8.5 ^a^	51.3 ± 7.9 ^a^	51.4 ± 9.5 ^a^
30	37.3 ± 7.0 ^a^	38.9 ± 7.0 ^a^	40.3 ± 9.2 ^a^	42.9 ± 4.9 ^a^	42.9 ± 7.5 ^a^

Lines with different letters denote significant different values at *p* < 0.01.

**Table 3 animals-10-00120-t003:** Recovery heart rate at different times after 60 min of exercise at different speeds (mean ± SD; 7 horses).

Time after Exercise (min)	Speed (m/s)	
	3.5	5
1	89.7 ± 8.9 ^a^	105 ± 11.7 ^b^
5	70.7 ± 22.2 ^a^	74.6 ± 8.9 ^a^
15	58.6 ± 12.6 ^a^	55.3 ± 8.5 ^a^
30	44.6 ± 16.1 ^a^	46.0 ± 9.7 ^a^

Lines with different letters denote significant different values at *p* < 0.05.

**Table 4 animals-10-00120-t004:** Time when the heart rate of 64 beats/minute was reached by the horses after exercise at different speeds for 10 and 60 min duration (mean ± SD; seven horses).

Exercise Speed	Exercise Duration	Time to Reach 64 beats/min
(m/s)	(min)	(s)
1.5	10	26 ± 51
3.5	10	75 ± 81
	60	194 ± 186
5	10	186 ± 126
	60	274 ± 198
6	10	367 ± 196
7	10	547 ± 297

**Table 5 animals-10-00120-t005:** Coefficient of determination (r^2^) and level of significance of the relationship between resting heart rate and recovery heart rate after 10 min of exercise at different speeds (seven horses).

Time (min)	Speed (m/s)				
	1.5	3.5	5	6	7
1	0.65 *	0.75 **	0.76 **	0.04	0.14
5	0.86 **	0.88 ***	0.86 **	0.29	0.18
15	0.40	0.87 **	0.64 *	0.52	0.40
30	0.62 *	0.90 ***	0.74 **	0.19	0.51
Time to reach 64 beats/min	0.49	0.54 *	0.87 **	0.32	0.18

* = *p* < 0.05; ** = *p* < 0.01; *** = *p* < 0.001.

**Table 6 animals-10-00120-t006:** Coefficient of determination (r^2^) and level of significance of the relationship between resting heart rate and recovery heart rate after 60 min of exercise at 3.5 m/s and 5 m/s (seven horses).

Time after Exercise (min)	Speed (m/s)	
	3.5	5
1	0.15	0.03
5	0.24	0.26
15	0.84 **	0.47
30	0.73 *	0.77 **
Time to reach 64 beats/min	0.02	0.00

* = *p* < 0.05; ** = *p* < 0.01.

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
