# Peer review of "Relationship between Resting and Recovery Heart Rate in Horses"

_animals, 2020, doi:10.3390/ani10010120_

Round 1

Reviewer 1 Report

Type of manuscript: Article
Title: Relationship between resting heart rate and recovery heart rate
in horses
Journal: Animals

The authors take up the subject of an application, but I do not understand why the escaped character referenced to endurance discipline

Since the research was conducted on a treadmill and in conditions different from the field conditions during an endurance, why the authors place such emphasis on this discipline, similar relationships occur in flat races. The obtained results can be used for horses participating in other equestrian competitions. Introduction and discussion should be extended about these issues.

Introduction

Introduction is quite poor, please add a development regarding horse physiology, performance tests. This part is too focused only on the endurance discipline.

Material and methods

The big problem is the varied age and experience of horses participating in the experiment. This factor can have a significant impact on horse HR after exercise

Were horses used to running on the treadmill before? If it could not have had a significant impact on the emotional state of horses, and thus on HR.

Discussion

Again, too much about endurance, and too little general information related to exercise physiology

P5 L159-171 this fragment would be better suited to the introduction

Author Response

The authors take up the subject of an application, but I do not understand why the escaped character referenced to endurance discipline

Our answer: HRrecovery plays an important role for horses during endurance races. This is not the case or to a much lesser extent for horses competing in other sports disciplines (in driving competitions the HR of horses is monitored carefully too, but does not play such a vital role for the horses to be allowed to continue the competition as for the endurance racing horses). Thus, the results of this study on the relationship between HRresting and HRrecovery play a practical role for horses competing in endurance racing only! This is the reason for the focus of the manuscript.

Since the research was conducted on a treadmill and in conditions different from the field conditions during an endurance, why the authors place such emphasis on this discipline, similar relationships occur in flat races. The obtained results can be used for horses participating in other equestrian competitions. Introduction and discussion should be extended about these issues.

Our answers

The results apply for all horses exercising, but have practical implications for horses competing in endurance racing only. We have changed the introduction and discussion somewhat, but do not consider it appropriate to expand the information on exercise physiology or testing in general and horses competing in other disciplines. We have done extensive literature searches to find that there seem not to be scientific publications describing the relationship between HRresting and HRrecovery. We would be very grateful, for the reference/s that you mention! Parallel to the study on the treadmill another group of horses underwent exactly the same study protocol under ridden conditions and on a sandy racing track. The results of this part of the study have not been published yet and probably have a reduced chance to be published only because very few relationships between HRresting and HRrecovery were found. Journals do not value this type of “negative” studies. We assume that the reason for the results of the outdoor study was the lower possibility to standardize the conditions compared with those of the treadmill study (outdoor, rider, track, other horses, etc.). We will try to publish those results too, but have decided to submit first the data presented in the actual manuscript.

Introduction

Introduction is quite poor, please add a development regarding horse physiology, performance tests. This part is too focused only on the endurance discipline.

Our answer: Correct, the content of the manuscript is focused on horses competing in endurance racing because of its medical practical implications. This is not the case for all other equine sport disciplines. This is the reason for not writing about horse physiology and performance testing in the introduction.

Material and methods

The big problem is the varied age and experience of horses participating in the experiment. This factor can have a significant impact on horse HR after exercise

Our answer: We discuss this. The low number of horses is certainly a limitation, We were lucky to have them, and with such a relatively large range of HRresting values!

Were horses used to running on the treadmill before? If it could not have had a significant impact on the emotional state of horses, and thus on HR.

Our answer: The horses were very well adapted to run on the treadmill!

Discussion

Again, too much about endurance, and too little general information related to exercise physiology

Our answer: We have tried to clarify why this is like we have written it.

P5 L159-171 this fragment would be better suited to the introduction

Our answer: We have added some passages of this paragraph to the introduction too.

Reviewer 2 Report

In my  opinion the paper is interesting because it focuses an important aspect of sport medicine in horses.

The discussion section needs to be reformulated just to make it simplier and easier to read. It is to verbose now.

Some grammar errors, thus it needs an editing for english language.

Author Response

The discussion section needs to be reformulated just to make it simplier and easier to read. It is to verbose now.

Our answer: We have made some deletions and modifications too, that hopefully will meet your suggestion!

Some grammar errors, thus it needs an editing for english language.

Our answer: Hopefully the editor/s can help out because we have read the manuscript again and carefully, but not found those grammar errors that you mention. The manuscript was revised by a professional language editing service before being submitted to Animals. For American English! We did not find an indication in the guidelines for submitting a paper that it should be British or another type of English.

Round 2

Reviewer 1 Report

In present form, the manuscript may be published in Animals